# A Commercial Carbonaceous Anode with a-Si Layers by Plasma Enhanced Chemical Vapor Deposition for Lithium Ion Batteries

**Chao-Yu Lee [1], Fa-Hsing Yeh [2] and Ing-Song Yu [2],***

[1]  Department of Materials Science and Engineering, National Formosa University, Yunlin 632, Taiwan; cyl@nfu.edu.tw

[2]  Department of Materials Science and Engineering, National Dong Hwa University, Hualien 974301, Taiwan; 610122027@ems.ndhu.edu.tw

[*]  Correspondence: isyu@gms.ndhu.edu.tw; Tel.: +886-3-890-3219

**Abstract:** In this study, we propose a mass production-able and low-cost method to fabricate the anodes of Li-ion battery. Carbonaceous anodes, integrated with thin amorphous silicon layers by plasma enhanced chemical vapor deposition, can improve the performance of specific capacity and coulombic efficiency for Li-ion battery. Three different thicknesses of a-Si layers (320, 640, and 960 nm), less than 0.1 wt% of anode electrode, were deposited on carbonaceous electrodes at low temperature 200 °C. Around 30 mg of a-Si by plasma enhanced chemical vapor deposition (PECVD) can improve the specific capacity ~42%, and keep coulombic efficiency of the half Li-ion cells higher than 85% after first cycle charge-discharge test. For the thirty cyclic performance and rate capability, capacitance retention can maintain above 96%. The thicker a-Si layers on carbon anodes, the better electrochemical performance of anodes with silicon-carbon composites we get. The traditional carbonaceous electrodes can be deposited a-Si layers easily by plasma enhanced chemical vapor deposition, which is a method with high potential for industrialization.

**Keywords:** lithium ion battery; silicon-carbon composites; anode; plasma enhanced chemical vapor deposition

## 1. Introduction

Lithium ion battery is the mainstream among the rechargeable batteries for consumer electronics. The demand of high power and high-energy density Li-ion battery is also increasing for the applications in electric vehicles and military. The commercial products of lithium ion batteries are mainly composed of three parts: anode, cathode and electrolyte. The cathode acts as the positive electrode of the battery during the discharge process, usually made with lithium metal oxide. The anode material is mostly composed of graphite or related carbon materials, and acts as the negative electrode during the discharge process.

So far, the carbonaceous materials and graphite with the low price are mainly used as the anode for the typical products of commercial Li-ion battery. Graphite can be regarded as a stack of hexagonally sheets of carbon, which is held together by van der Waals forces. The forces between two parallel carbon sheets are much weaker than the two given carbons (covalent bond) in the same sheet. Therefore, Li ions ($Li^+$) are able to insert in between the planes of graphite. This process is the mechanism, known as insertion or intercalation, to store lithium atom in graphite. During intercalation, lithium occupies an interstitial site between two planes of graphite and prevents other lithium ions to bind in adjacent interstitial sites. In other words, that means every six carbon atoms can only be combined with one lithium ion. The structure of graphite sheet is suitable for intercalation and de-intercalation of Li atoms

during charge and discharge process, which provides the high stability and reliability for the products of Li-ion battery. However, this is directly limited the energy density of graphite anode for Li-ion batteries. The capacity of graphite anode has a low theoretical limiting value as 372 (mA·h·g$^{-1}$) [1–3].

To improve the capacity of Li-ion battery, another candidate material, silicon, was proposed. Crystalline silicon can react with lithium to form a series of intermetallic compounds. Among them, Li$_{22}$Si$_5$, theoretical specific capacity 4200 mA·h·g$^{-1}$; is considered the most potential candidate to meet the demand, but it has suffered from the large volume expansion (>400%) during the lithiation and de-lithiation cycling. Such a large volume change will cause the crack and damage the structure of silicon electrodes, which induce huge capacity loss and safety issue. The poor electrical conductivity of silicon also reduces its effectiveness as an anode electrode of Li-ion battery. Those problems of crystalline Si anode electrode can result in the low coulombic efficiency and poor cycling stability [4–6]. To avoid these issues, silicon on coppers in the form of thin film, microwire and nanotube as anode electrode were proposed and investigated [7]. Besides, amorphous silicon electrode has better capacity retention and cycle life due to the presence of defects and absence of long range order, the volume expansion upon lithium insertion is believed to be less severe [8].

Moreover, to combine the advantage of carbon and silicon materials for the anode of Li-ion batteries, various carbon-silicon composite materials have been proposed for the high performance anode of Li-ion battery [9,10]. For instance, micro-spherical Si-C composite with hierarchical core-shell structure were prepared by spray drying polyvinyl alcohol solution containing silicon nanoparticles and followed by surface coating polycrylonitrile and heat treatment [11]. The surface of graphite powder with Si was prepared by mechano-chemical milling technique [12]. Dispersing Si in the matrix of carbon proved to be an effective method to solve the problem of expansion fracture. Using carbon as a matrix not only provides capacity and increases electrical conductivity, but also provides a ductile host matrix for dispersed Si particles. As a result, Si-C composite materials show higher capacity than carbonaceous materials, and have better cycle stability than Si electrodes. For the matrix materials, different types of carbon materials can be used, such as natural graphite, artificial graphite, carbon-nanotubes and Graphene [13,14]. So far, various technologies are employed to fabricate Si-C composite anodes such as mechanical milling, chemical vapor decomposition, magnetic sputtering and spray-pyrolysis approaches or some combination of these two to make Si particles embedded in a carbonaceous matrix [15,16]. Amorphous silicon–carbon composites synthesized by spray-pyrolysis approach from organic silane precursors with better mechanical buffering and excellent coulombic efficiency was observed when the amorphous silicon is uniformly dispersed and well embedded in the carbon matrix [17]. A nanosized Si/graphene composite was prepared by magnesium thermal reduction, which delivered an initial reversible capacity in the value of 1750 mA·h·g$^{-1}$ and excellent cycling stability with a capacity cycling stability with a capacity as 1374 mA·h·g$^{-1}$ after 120 cycling test [18]. The C-Si composites can be in the various forms of mixtures, inter-dispersions, porous, C coated by Si, Si coated by C, or core-shell structures [19,20]. Porous C-Si composite particles may also be a useful solution to overcome the limitations of dense C-Si electrodes. Pores in C-Si composite provide the volume needed for Si expansion and electrolyte in open pores allow Li ions to fast transport [21]. The nano-Si particles embedded in a shapeless poly(3,4-ethylenedioxythiophene):poly(styrenesulfonate) matrix and amorphous carbon was proposed for the anode material of Li-ion battery [22]. However, synthesis of such composites by a low-cost process is still an arduous task and challenge.

Coating technique, an easier way for the process integration, was also employed in Li-ion batteries to enhance their performance. For instance, atomic layer deposition (ALD) technique was conducted to fabricate thin Al$_2$O$_3$ layer on the surface of a polyvinylidene fluoride membrane to obtain the higher thermal stability, better affinity with electrolytes, higher ion conductivity and better Young's modulus [23]. For the anode electrodes, amorphous Si/C multilayer films by magnetic sputtering process as anode materials for lithium ion batteries. The micro-scale Si/C multilayer films showed a better long cycling performance [24]. Amorphous and nanocrystalline silicon was prepared on graphitic carbon fibers as Li-interclation materials by microwave plasma chemical vapor deposition

(CVD). More than 700 mAh/g capacity was obtained for the C-Si composite with silicon content around 20 wt.% [25]. The preparations of micro-scale Si/C multilayer and high Si content on carbon fibers required the long deposition time, which increased the difficulty of the mass-production. The processing temperature was usually up to several hundreds of degrees Celsius, which also limited the electrode materials used and raised the cost of the fabrication.

These C-Si composite anodes have demonstrated the great performance with the high specific capacity and excellent capacity retention for Li-ion batteries. However, these techniques still has some drawbacks for the industrial applications because of the high-cost, time-consuming and low-possibility for the mass production of high performance Li-ion batteries. Plasma enhanced chemical vapor deposition (PECVD) technique, used for the mass-fabrication of photovoltaic industry, deposits silicon nitride ($SiN_x$) thin films as the antireflection coating layer and as the hydrogenatated passivation layer for crystalline silicon solar cells [26–28]. It can also be employed for the deposition of amorphous, nano-crystalline or micro-crystalline silicon in the application of thin film silicon solar cells [29]. The PECVD method can be conducted at very low deposition temperature and provide a roll-to-roll process for mass production. Therefore, this technique could be a low-cost and mass production-able method for the anodes of Li-ion batteries.

In this study, we propose an effective method to fabricate Si-C composite anodes: the commercial carbonaceous anodes deposited amorphous-Si layers by PECVD. PECVD technique has some advantages, such as simple structure, low material consumption, high throughput, low process temperature, and low cost of production. The process integration of traditional carbon anode with a-Si layers can be realized easily in the manufacture of Li-ion battery. These Si-C composite anodes were investigated as anode of Li-ion batteries and exhibited a better electrochemical performance than the traditional carbon anode. The characterizations of our Si-C composite electrodes were conducted after PECVD growth of a-Si and the electrochemical cycling test.

## 2. Materials and Methods

For the preparation of Si-C composite anode for Li-ion battery, Figure 1 shows the schematic diagram of CR2030-type coin cells. The mesocarbon microbeads (MCMB) graphite powders were mixed with two conducting media (Super-P carbon black and KS-4 synthetic graphite) and a binder (polyvinylidene fluoride, PVDF) as the weight ratio of 80:5:5:10 in N-Methyl-2-pyrrolidone (NMP) solvent. The mixture was stirred by a ball mill for 12 h to prepare the uniform electrode slurry. Then, the electrode slurry was pasted on a 10 μm-thick copper foil substrate by using a doctor blade, and put in a blow dryer to evaporate the solvent. The prepared anode sheets were dried at 135 °C in a vacuum oven for 12 h. After cooling to room temperature, the sheets were pressed under a pressure of approximately 200 kg·cm$^{-2}$ to regulate the thickness of electrode layers about 100 μm. Finally, the a-Si layer had been deposited on the electrode sheets by the PECVD technique. For the PECVD process, the parameters were $SiH_4$ as precursor gas with flux 28 sccm, the power 90 W, the pressure 0.64 torr, and the temperature of 200 °C. In this work, three different thicknesses (320, 640 and 960 nm) of a-Si layers were deposited on carbon anodes at the deposition rate of 40 nm/min. The thickness of a-Si layers was identified on the Si wafers because of the rough carbon electrode. To characterize the a-Si/C compostite anodes, we employed scanning electron microscopy (SEM) for the surface morphology observation, electron probe micro-analysis (EPMA) and Raman spectroscopy for the analysis of chemical composition.

For the electrochemical analysis, CR2032-type coin cells were used to measure electrochemical performance of one traditional carbon anode and three a-Si/C composite anodes. A lithium foil and a porous polypropylene film were served as the counter electrode and the separator, respectively. The electrolyte solution was 1.0 M LiPF6 in a mixture of ethylene carbonate, propylene carbonate, and dimethyl carbonate with a weight ratio of 1:1:1. The coin cells were assembled in an atmosphere controlled glove box. The charge/discharge cycling test at different C rates (from 0.1 to 5 C) was conducted with the voltage from 0.01 to 1.2 V versus Li$^+$/Li at the ambient temperature.

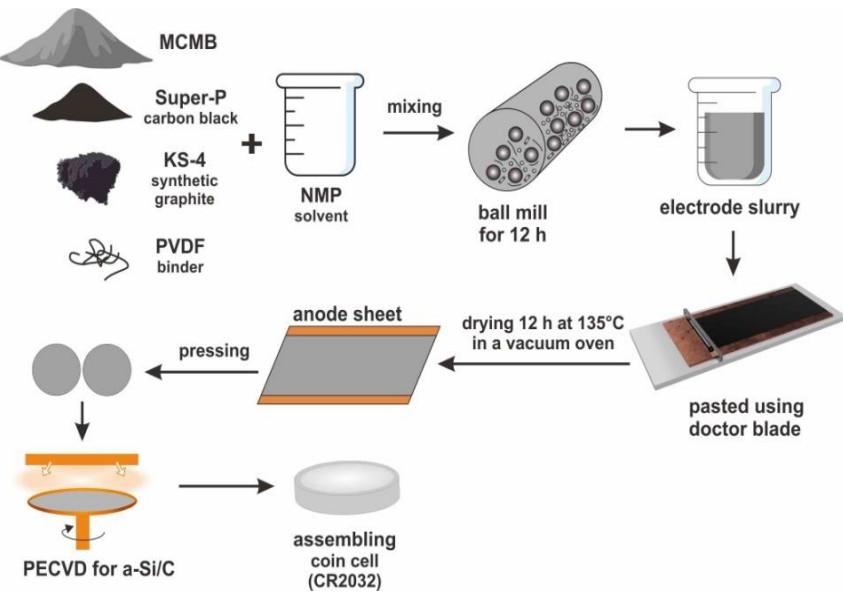

**Figure 1.** Schematic diagram for the preparation of CR2032-type coin cells.

## 3. Results and Discussion

Figure 2a,b show the surface morphology of carbonaceous anodes with and without a-Si layers coating, respectively. The carbonaceous anode was composed by big MCMB graphite, sheet KS-4 in around 5 μm, and spherical Super-P in around 40 nm. After a 320 nm-thick a-Si layer by PECVD, the surface was covered by Si, and the edge of flake graphite became unclear. In Figure 2c, energy-dispersive X-ray spectroscopy (EDX) was conducted at the point of a MCMB graphite as shown in Figure 1b. The chemical composition included 66.43% of carbon, 27.21% of silicon, 3.83% of oxygen, and 2.53% of fluorine after deposition. Si was deposited on the carbon anode by PECVD, and fluorine came from the binder PVDF.

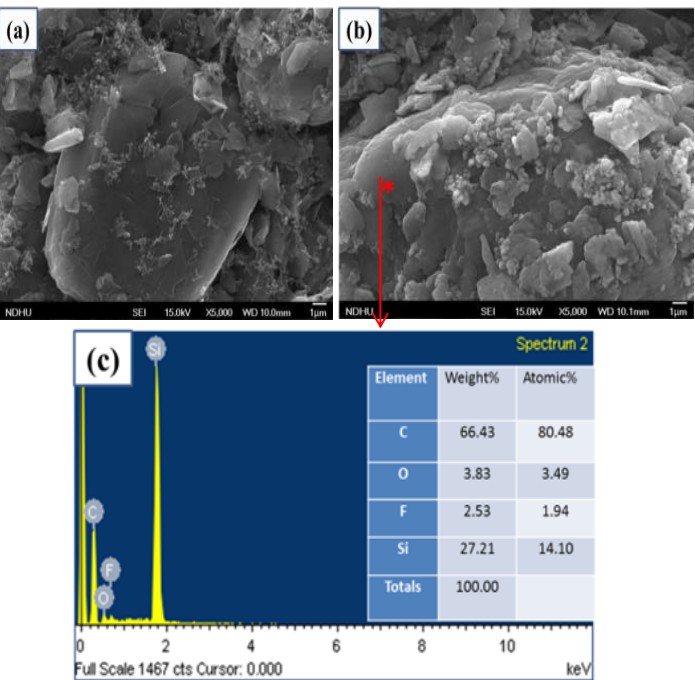

**Figure 2.** SEM images of carbonaceous anodes, 5000×: (**a**) Without a-Si coating, (**b**) With 320 nm-thick a-Si coating, and (**c**) EDX spectrum of a-Si/C composite anode.

To further investigate the coverage of a-Si on carbon electrode, the mappings of EPMA were used to determine the concentrations of the constituent elements such as copper, carbon and silicon in the sample extracted from the anode electrodes with different thicknesses of a-Si layers. Figure 3 shows the EPMA mapping results in 0.2 mm × 0.2 mm area on the carbonaceous anodes with 320 nm-thick and 960 nm-thick a-Si layers. A few Cu signals came from the substrate of electrodes, which means most of the region covered by the a-Si/C composite. In Figure 3a, for the electrode with a 320 nm-thick a-Si layer, signals of carbon are stronger than those of silicon, especially for the boundary of MCMB graphite. On the other hand, for the electrode with a 960 nm-thick a-Si layer in Figure 3b, signals of silicon became stronger, which indicates a-Si layer became thicker and covered MCMB anode more completely. From the results of EPMA, the rough surface of carbon anode can be uniformly covered with a-Si by PECVD technique, except for the recess of carbon anode.

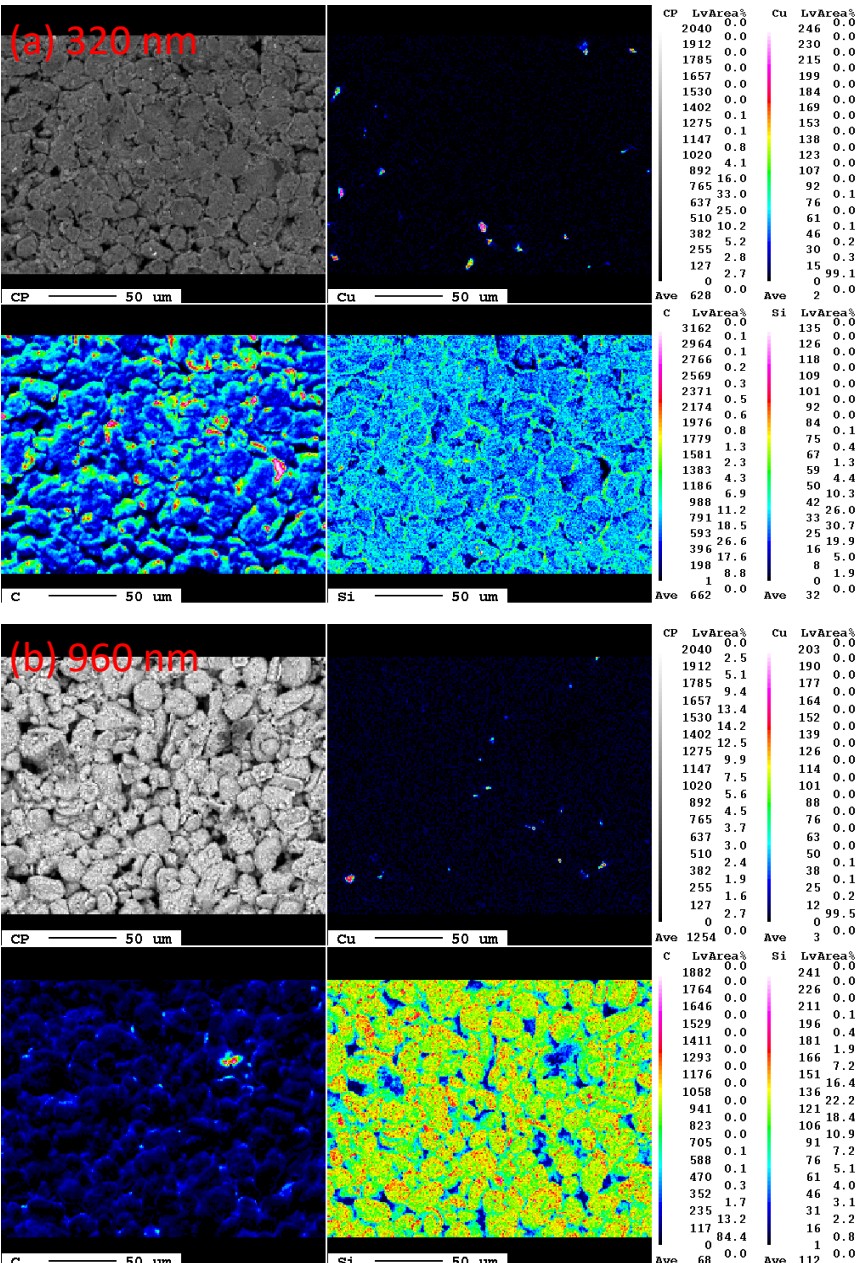

**Figure 3.** (**a**) EPMA mapping of the carbonaceous anode with a 320 nm-thick a-Si layer, and (**b**) EPMA mapping of the carbonaceous anode with a 960 nm-thick a-Si layer.

For the electrochemical performance of anodes, Figure 4a shows the typical charge-discharge profiles of a carbon electrode and three a-Si/C composite electrodes in the first cycle. The first discharge capacity of electrode with 960 nm-thick a-Si layer (green line) was 572.32 mA·h·g$^{-1}$, and the reversible charge capacity was 487.77 mA·h·g$^{-1}$. To compare with the carbon electrode, 42% of capacity was improved due to around 30 mg of a-Si adding by PECVD. A small amount of a-Si additives has a great improve on the discharge capacity. This is because the theoretical capacitance value of silicon has more than ten-time higher than that of carbon [30–32]. In addition, amorphous silicon nano-flake provides more location for lithium ion to insert and extract than granular silicon crystal, therefore, in this report, a small amount of a-Si addition can achieve a significant enhancement of capacity.

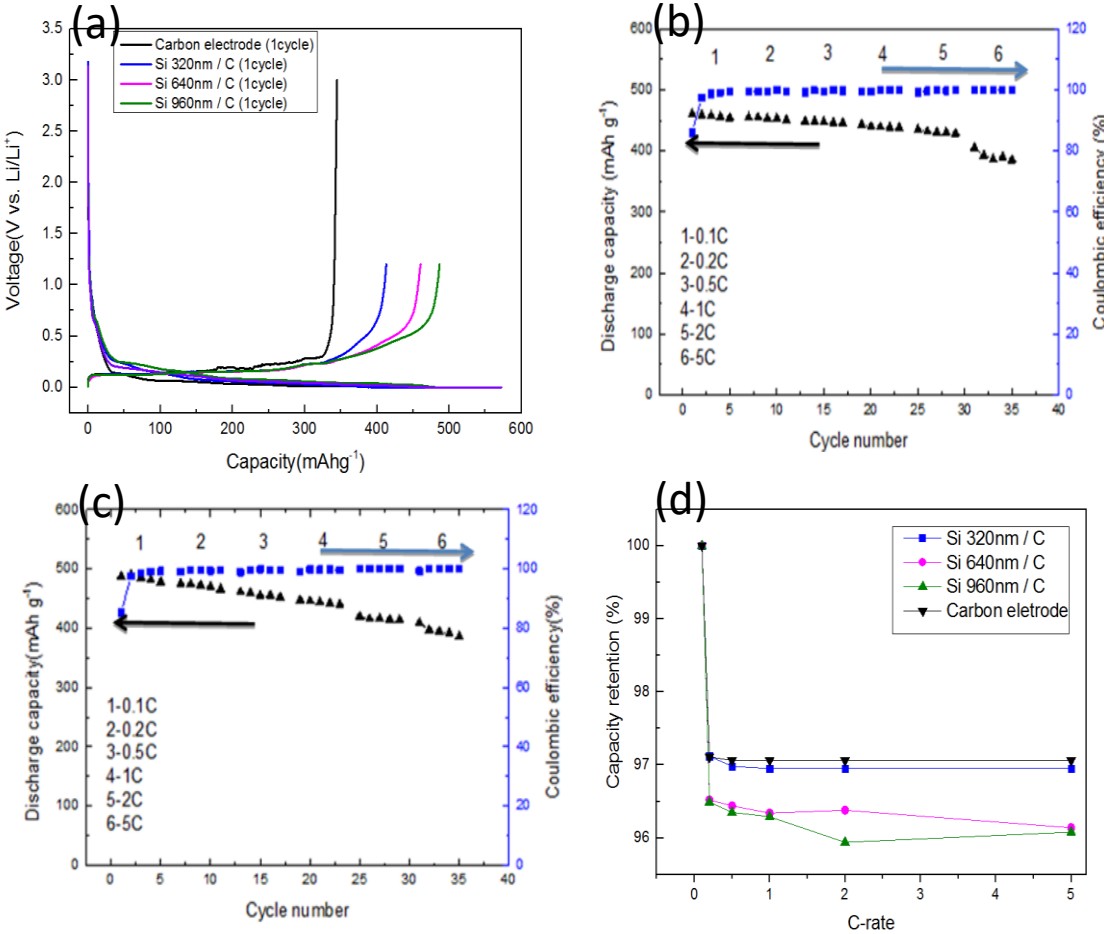

**Figure 4.** (**a**) First charge-discharge test of one carbon and three a-Si/C composite electrodes, (**b**) Charge-discharge test at different rates for the a-Si(640 nm)/C composite electrode, (**c**) Charge-discharge test at different rates for the a-Si(960 nm)/C composite electrode, and (**d**) Capacity retention ratio for all the electrodes.

The first discharge capacity of half cells increased with the thickness of a-Si layers. The coulombic efficiencies in first cycle for all a-Si/C composite electrodes can keep as high as 85%, which shows the high cycling stability of a-Si/C composite anode for Li-ion batteries. The coulombic efficiencies in first cycle are higher than other low-cost process such as electrodeposition of amorphous silicon anode [33], that can be explained by nanostructured and amorphous silicon flakes on carbonaceous anodes. Because the base anode was composed by MCMB and sheet graphite that means the surface of the anode will not be totally smooth, coated a-Si will not able to form a complete film, which can only form silicon flakes attached to the surface of graphite. The coated silicon with nanostructured and

amorphous forms provide mechanical integrity without broken due to the nano-sized grain and the free expansion in amorphous silicon which makes better capacity retention and cycle life of lithium battery.

　　For the cyclic performance and rate capability, half cells were activated initially at 0.1 C for 5 cycles and then tested at different rates as 0.2 C, 0.5 C, 1 C, 2 C and 5 C during the prolonged cycling. Figure 4b shows the discharge capacity and coulombic efficiency of the anode with a 640 nm-thick a-Si layer. The discharge capacity of a-Si/C electrode is about 455 mA·h·g$^{-1}$ at a rate of 0.1 C and remains more than 380 mA·h·g$^{-1}$ at 5 C. The discharge capacity of the a-Si/C electrode did not decade significantly until it was tested at 5 C. After the first cycle, the a-Si/C anode shows highly reversible behavior and coulombic efficiencies for a-Si/C cells are more than 98% for the other cycling tests. The stable coulombic efficiency of a-Si/C composite electrodes could be attributed to the free expansion of a-Si layers on the carbon electrodes.

　　Figure 4c shows the discharge capacity and coulombic efficiency of the electrode with a 960 nm-thick a-Si layer. The capacity of a-Si/C was about 490 mA·h·g$^{-1}$ at 0.1 C and remained more than 390 mA·h·g$^{-1}$ at 5 C. The discharge capacity of a-Si/C electrodes increased with the thickness of a-Si layer. Although the capacity value gradually declined with the increase of charge-discharge rates, the stable coulombic efficiency of a-Si/C anode performed. Small amounts of a-Si coated by PECVD on carbon electrodes can improve the electrochemical performance of the anode for Li-ion battery.

　　Figure 4d shows capacity retention of a carbon electrode and three a-Si/C composite electrodes at different C rates. The capacity retention ratio between five charge-discharge cycles retained more than 97% for carbon electrode and a-Si/C electrode with 320 nm-thick Si. The difference of retention ratio between carbon electrode and the one with 320 nm a-Si layer is tiny, because the amount of additives is small, so that the effect of the carbon electrode plays a major role. However, as the thickness of a-Si layer increased, capacity retention ratio slightly decreased. For the electrodes with thicker a-Si layers, capacity retention reduced to around 96% for different C-rate. Compared with the Si electrode, this good capacity retention of a-Si/C anodes could be attributed to strong adhesion between a-Si and MCMB graphite.

　　For the further discussion on the electrochemical performance of a-Si/C electrodes, the higher specific capacities of a-Si/C electrodes come from the small amount a-Si on the surface of carbon electrode by PECVD. From the observation of surface morphology and chemical composition, a-Si layers were deposited onto the extremely rough surface of carbon electrodes. The deformation of Si between lithiation and delithiation could not been constrained on the rough surface to avoid mechanical disintegration of the electrodes [34]. Our rough electrodes provided a free space for a-Si layer expansion and could prevent internal stress occurred and improve mechanical durability of a-Si/C electrodes. Therefore, the a-Si/C electrodes show high cycling stability and coulombic efficiency during the electrochemical test. In addition, excellent adhesion of a-Si on carbon performs good capacity retention. For the thicker a-Si on carbon electrode, the capacity retention decreased. This is because of the increased Li diffusion length, higher electrical resistance and larger internal stress of Li insertion/extraction.

　　Raman spectroscopy was also used to investigate a-Si/C electrodes. Figure 5 shows Raman spectra for four samples. Carbon electrode (black line) included two typical Raman vibration modes, D band around 1350 cm$^{-1}$ and G band around 1590 cm$^{-1}$. After the a-Si deposition, the a-Si/C electrode (red line) had the main Si peak around 520 cm$^{-1}$, and the D and G bands of carbon disappeared, which means the surface of carbon electrode was covered by a-Si. After the charge-discharge test for 30 cycles, the D and G bands of carbon appeared obviously for the electrode with the thinner a-Si layer (pink line). This could be because of some plastic deformation of Si during lithiation and delithiation processes. D and G bands of carbon still did not be observed for the electrodes with thicker a-Si layer (green line), which means the carbon electrode was better protected by thicker a-Si layers.

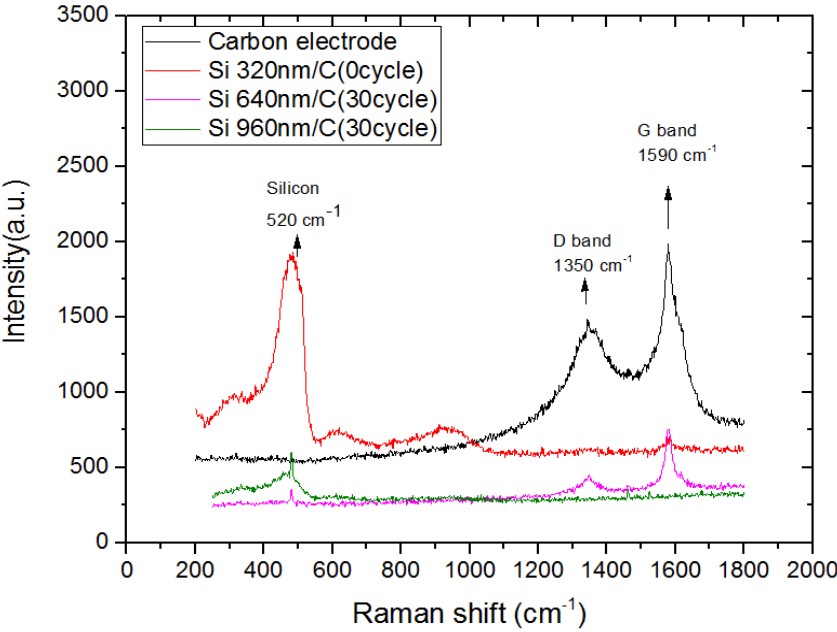

**Figure 5.** Raman spectra of one carbon electrode (black), one as-prepared a-Si/C electrode (red), and two a-Si/C electrodes after charge-discharge test for 30 cycles.

In general, the cycled test of carbon anode electrode shows the larger porosity, smaller tortuosity and similar specific surface area compared to the new electrode for the geometric characterization [35]. Here, the surface morphology of a-Si/C electrodes was observed after the charge-discharge test for 30 cycles. Figure 6 shows the SEM images of two a-Si/C electrodes with 640 nm and 960 nm a-Si layers. The electrodes did not have large volumetric strain to cause mechanical damage, but local crack still could be found. The thicker a-Si layer made the electrode more complete after the cycling test, which was consistent with the result of Raman spectra. The rough carbon electrodes could provide good surface adhesion and free space expansion for a-Si layers, which can improve the performance and durability of a-Si/C composite electrodes.

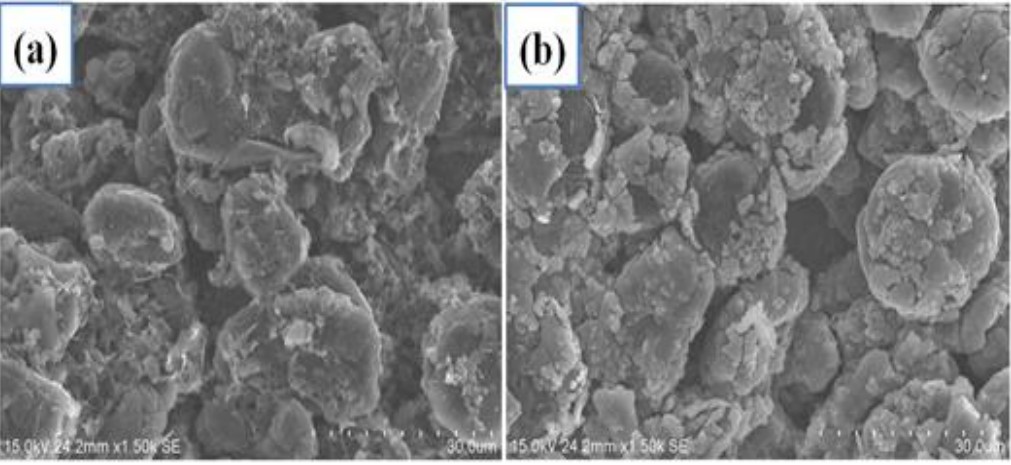

**Figure 6.** SEM images of two a-Si/C electrodes after the charge-discharge test for 30 cycles, 1500×: (**a**) Electrode with a 640 nm a-Si, and (**b**) Electrode with a 960 nm a-Si.

The effect of small addition of silicon on the performance of the carbon electrode was studied; the tiny amount of a-Si was used because if the theoretical capacity of the negative electrode active

material is increased much, the corresponding positive electrode active material formula also needs to be greatly changed for the commercial Li-ion battery. The main consideration in this study is that based on the current process of lithium-ion batteries in the industry, the overall performance of lithium-ion batteries can be improved easily by PECVD a-Si process. Compared with the negative electrode active materials with high-capacity published in the literatures [36–38], the increasing in the capacitance of this study is not significant. However, high-capacity active materials have their use restrictions, especially for the reliability and process integration for mass-production. In the work, we proposed a possible solution for the industrial Li-ion battery. For instance, PECVD technique is a fabricating method used in photovoltaic products, which that has been proven to be capable of quantitative production. It is also adopted by most companies because of its lower cost advantage. In addition, only a very small amount of silicon is added in this study, no matter the consumption of materials and the energy consumption of the process are very small, it should theoretically be able to meet the low-cost requirements.

## 4. Conclusions

We proposed a new mass production-able and low-cost fabrication method of a-Si/C composite anode for Li-ion battery: traditional carbonaceous electrodes deposited a-Si layers by PECVD at low temperature. PECVD process is widely adopted in the solar panel industry because of its low cost advantage. This idea combines the advantage of carbon electrode in stability, reliability and mass production, as well as the merit of Si in high theoretical specific capacity. Small amounts a-Si on the rough surface of carbon electrodes can improve obviously the electrochemical performance of the carbon anode such as specific capacity and coulombic efficiency without much material consumption. For the a-Si deposition thickness of 960 nm, 42% of capacity can be improved due to only around 30 mg of a-Si deposited by PECVD. The more a-Si on the carbon electrode, the higher capacity of the anode we get. These depositions (320 nm, 640 nm and 960 nm) can avoid the pulverization issue of Si electrode for the Li-ion battery. Coulombic efficiencies of a-Si/C electrodes are very stable and greater than 85% after the first charge-discharge test. Capacitance retention ratio of a-Si/C electrodes can be maintained 96% at different C rates. Amorphous and flake like silicon allow free expansion during limitation and de-limitation, which makes better capacity retention and cycle life. The rough carbon electrodes also can provide the good adhesion and free expansion for a-Si layers, so a-Si/C composite anodes have better electrochemical performance than traditional carbonaceous electrodes. The carbon anode with a-Si layers by PECVD has high potential for the manufacture of Li-ion battery.

## 5. Patents

Invention patent of Taiwan, patent title: Lithium ion battery and composite electrode material thereof, and fabrication method of composite electrode material, patent number: I584519.

**Author Contributions:** Conceptualization, C.-Y.L.; methodology, I.-S.Y.; data curation, F.-H.Y.; writing—original draft preparation, I.-S.Y., and C.-Y.L. All authors have read and agreed to the published version of the manuscript.

**Funding:** This research was funded by Ministry of Science and Technology, Taiwan. Grant number: 103-2623-E-259-001-D and 107-2221-E-259-001-MY2.

**Acknowledgments:** Authors acknowledge the researchers in Chun-Shan Institute of Science and Technology Po-Yen Chen and Bi-Sheng Jang for the help in experiment and discussion.

**Conflicts of Interest:** The authors declare no conflict of interest.

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
