# Peer review of "A Commercial Carbonaceous Anode with a-Si Layers by Plasma Enhanced Chemical Vapor Deposition for Lithium Ion Batteries"

_jcs, doi:10.3390/jcs4020072_

Round 1

Reviewer 1 Report

This work mainly introduces the a-Si layers by Plasma enhanced CVD that is coated on the surface of carbonaceous anode for LIB with better performance. The work is interesting and well-written. The capacity and cycling stability of Si/C anode based half cells are greatly improved. The corresponding mechanism is also investigated. So, I suggest this work can be accepted after minor revision.

  • After the coating of a-Si layer, if did this procedure takes any reactive materials into the anode system. Authors should provide the first cycle Coulombic efficiency for all samples and provide a related explanation.
  • In the case of Figure 3a, the capacity is obviously improved, it is so amazing, the author should provide an explanation in detail.
  • As we know, coating techniques are widely employed in LIB for better performance. Some other works on this should be introduced as a reference and cited such as ACS Applied Energy Materials 2 (6), 4167-4174.
  • The capacity vs cycles files of various cells under different C-rate should be presented, not just the conclusion like Figure 3d.
  • If possible, the chemical stability of modified anode should be tested through CV.

Reviewer 2 Report

Lee et al. introduced a Commercial Carbonaceous Anode with a-Si Layers for Lithium Ion Batteries. They provided the detail of the fabrication with adequate evidence. The manuscript can be improved by minor revision.

  • Providing a scheme that illustrates the fabrication process.
  • Add more recent articles in the field and compare the performance of the fabricated anode with the previously reported and/or commercially available.
  • Provide an estimated cost of the designed electrode.
  • The conclusion is almost the same as abstract; I suggested to rewrite it to be more strength.
  • Some suggestions for the citation:

Graphene caging silicon nanoparticles anchored on graphene sheets for high performance Li- ion batteries. Applied Surface Science 484 (2019) 11–20

Nano-ordered structure regulation in delithiated Si anode triggered by homogeneous and stable Li-ion diffusion at the interface. Nano Energy 72 (2020) 104651

Nanoporous SiOx coated amorphous silicon anode material with robust mechanical behavior for high-performance rechargeable Li-ion batteries. Nano Materials   Science 1 (2019) 70–76

Silicon nanofilms as anode materials for flexible lithium ion batteries. Thin Solid Films 690 (2019) 137516

Reviewer 3 Report

The manuscript describes PECVD derived a-Si on carbonaceous anode and studies the LIB performance.

It’s not clear how the thickness of a-Si is controlled and measured. The authors should explain better. Three thicknesses were studied including 320, 640 and 960 nm. How about the electrochemical behavior for samples with a thickness over 960 nm?

Comparison of the capacity with previous reports is required. The authors should explain why the capacities obtained are so low (much lower than 100 mAh g-1).

In Fig. 3b-c, the LIBs should be tested at 0.1C after 5C.

Round 2

Reviewer 3 Report

The reviewer is satisfied with the revision and would like to recommend the acceptance of the manuscript.